# Initial Experience of Atezolizumab Plus Bevacizumab for Unresectable Hepatocellular Carcinoma in Real-World Clinical Practice

**DOI:** 10.3390/cancers13112786

**Published:** 2021-06-03

**Authors:** Hideki Iwamoto, Shigeo Shimose, Yu Noda, Tomotake Shirono, Takashi Niizeki, Masahito Nakano, Shusuke Okamura, Naoki Kamachi, Hiroyuki Suzuki, Miwa Sakai, Akira Kajiwara, Satoshi Itano, Masatoshi Tanaka, Taizo Yamaguchi, Ryoko Kuromatsu, Hironori Koga, Takuji Torimura

**Affiliations:** 1Division of Gastroenterology, Department of Medicine, Kurume University School of Medicine, Kurume 830-0011, Japan; shimose_shigeo@med.kurume-u.ac.jp (S.S.); noda_yuu@med.kurume-u.ac.jp (Y.N.); shirono_tomotake@med.kurume-u.ac.jp (T.S.); niizeki_takashi@med.kurume-u.ac.jp (T.N.); nakano_masahito@med.kurume-u.ac.jp (M.N.); okamura_shyuusuke@kurume-u.ac.jp (S.O.); kamachi_naoki@med.kurume-u.ac.jp (N.K.); suzuki_hiroyuki@med.kurume-u.ac.jp (H.S.); sakai_miwa@med.kurume-u.ac.jp (M.S.); kajiwara_akira@med.kurume-u.ac.jp (A.K.); ryoko@med.kurume-u.ac.jp (R.K.); hirokoga@med.kurume-u.ac.jp (H.K.); tori@med.kurume-u.ac.jp (T.T.); 2Department of Gastroenterology, Iwamoto Internal Medicine Clinic, Kitakyushu 802-0832, Japan; iwamotos@orion.ocn.ne.jp; 3Department of Gastroenterology, Kurume Central Hospital, Kurume 830-0001, Japan; angiolion@h9.dion.ne.jp; 4Department of Gastroenterology, Yokokura Hospital, Miyama 839-0295, Japan; mazzo6528@me.com

**Keywords:** hepatocellular carcinoma, atezolizumab, bevacizumab, real-world data

## Abstract

**Simple Summary:**

Although the clinical trial of atezolizumab plus bevacizumab have revealed its efficacy for HCC, the outcomes in the real-world clinical practice are unclear. In the study, we retorspectively evaluated the efficacy and safety of atezoizumab plus bevacizumab for HCC. Atezorizumab plus bevacizumab was effective and safe even in the real-world clinical practice including patients with HCC in a previous MTA history or other than ALBI grade 1.

**Abstract:**

Background: Atezolizumab plus bevacizumab was approved for patients with hepatocellular carcinoma (HCC). Although clinical trials have revealed its efficacy, the outcomes in the real-world clinical practice are unclear. We retrospectively evaluated the efficacy and safety of atezolizumab plus bevacizumab for HCC. Materials and Methods: This is a multicenter study conducted between November 2020 and March 2021. Among the 61 patients, 51 were assessed for progression-free survival (PFS), therapeutic response, and adverse events (AEs). Results: The median PFS was 5.4 months. The objective response rate (ORR) was 35.3%. The disease control rate (DCR) was 86.3%. The incidence rates of AEs at any grade and grade >3 were 98.0% and 29.4%, respectively. The most frequent AE at any grade and grade >3 was hepatic disorder. In patients with a previous history of molecular targeted agent (MTA) or the degree of albumin-bilirubin (ALBI) grade, there were no significant differences in the PFS, ORR, DCR, and incidence rates of AEs. Conclusion: The study demonstrated that atezolizumab plus bevacizumab was effective and safe for patients with HCC even in the real-world setting including patients with a previous MTA history or other than ALBI grade 1.

## 1. Introduction

Sorafenib was the only approved systemic treatment for unresectable hepatocellular carcinoma (HCC) 10 years ago [1]. However, systemic treatment for HCC remarkably improved as options for first-line and second-line systemic treatments increased [2,3]. Previous molecular targeted agents (MTAs) such as anti-angiogenic drug (AAD) mainly targeted tumor angiogenesis [4]. The recent trend is a combination therapy of immune check point inhibitor (ICI) with AAD [5]. ICI is a cancer immunotherapy that has been increasingly used for various cancer types [6]. Representative target molecules are programmed cell death 1 (PD-1), programmed death-ligand 1 (PD-L1), and cytotoxic T-lymphocyte associated protein 4 (CTLA-4) [7]. 

The combination therapy of atezolizumab, a monoclonal antibody targeting PD-L1, and bevacizumab, a monoclonal antibody, against vascular endothelial growth factor (VEGF) A significantly prolonged progression-free survival (PFS) and overall survival (OS) in patients with unresectable HCC in the IMbrave 150 phase 3 clinical trial [8]. According to the results of this clinical trial, the first-line therapy in the systemic treatment of advanced HCC dramatically shifted from the oral tyrosine kinase inhibitors, such as sorafenib and lenvatinib, to atezolizumab plus bevacizumab [9,10]. All enrolled cases were MTA-naïve and all cases were Child-Pugh (C-P) Class A. Afterwards, the combination therapy revealed significantly better therapeutic outcomes and safety than sorafenib. In real-world clinical practice, the combination therapy can be used not only for the MTA-naïve cases but also the MTA-experienced cases as second- or third-line treatment [11]. Therefore, clarification is needed whether atezolizumab plus bevacizumab can be administered even for the MTA-experienced cases with safety. Additionally, the detailed evaluation regarding adverse events (AEs) is also needed. 

In this study, we analyzed overall therapeutic outcomes and safety of the initial experience of atezolizumab plus bevacizumab for unresectable HCC. Especially, we focused on the therapeutic outcomes and safety in the difference of MTA-naïve or -experienced cases and in the difference of albumin-bilirubin (ALBI) grade.

## 2. Materials and Methods

### 2.1. Study Design and Patients

This study retrospectively evaluated 61 patients with unresectable HCC who were treated with atezolizumab plus bevacizumab between November 2020 and March 2021 at four independent institutions (Kurume Liver Cancer Study Group of Japan) in Japan. The following inclusion criteria were used: (i) HCC diagnosed by radiological evaluation using enhanced computed tomography (CT) or magnetic resonance imaging (MRI) combined with serum tests for tumor markers; (ii) age >18 years; and (iii) complete follow-up from the initial treatment until death or the study censor time (31 April 2021). The exclusion criteria included the following: cases from which data could not be collected precisely and (ii) cases that were followed up only for a short period (<2 weeks). 

Clinical characteristics, therapeutic responses including PFS and radiological findings, and AEs were analyzed. From the medical records of the enrolled patients, patient information including sex, age, HCC etiology, C-P class, alpha-fetoprotein (AFP), des-γ-carboxy prothrombin (DCP), tumor size, and previous MTAs history was collected. HCC was classified using the Barcelona Clinic Liver Cancer (BCLC) staging system and TNM staging of the Liver Cancer Study Group of Japan. Additionally, albumin-bilirubin (ALBI) grade was calculated based on serum albumin and total bilirubin values according to the following formula: [ALBI score = (log10 bilirubin (μmol/L) × 0.66 + (albumin (g/L) × −0.085)], and defined by the following scores: ≤−2.60 = Grade 1, > −2.60 to ≤ −1.39 = Grade 2, >−1.39 = Grade 3 [12].

This study was conducted according to the Helsinki Declaration and approved from the ethics committee of Kurume University School of Medicine (approval number: 20183). Written informed consent was collected from each patient. However, this was waived because of the retrospective study design.

### 2.2. Treatment Protocol

The combination therapy of atezolizumab plus bevacizumab (Chugai Pharmaceutical Co. LTD, Tokyo, Japan) was performed according to the pharmaceutical recommendation. Patients received 1200 mg of atezolizumab plus 15 mg/kg of bevacizumab intravenously every 3 weeks. Patients received the treatment until development of unacceptable AEs or tumor progression. Treatments could continue even beyond tumor progression if the clinical benefit was observed. Patients who developed AEs were allowed to continue receiving a monotherapy of either atezolizumab or bevacizumab according to the developed AE profile.

### 2.3. Evaluation of Therapeutic Response and Follow-up Schedule

Tumors were assessed by CT or MRI every six weeks after initiation of treatment. The therapeutic response was evaluated according to the Response Evaluation Criteria in Solid Tumors (RECIST) [13] and the modified RECIST guidelines [14]. Objective response rate (ORR) was assessed as complete response (CR) + partial response (PR). Disease control rate (DCR) was assessed as ORR + stable disease (SD). In evaluation of the therapeutic response, two independent hepatologists assessed the patients in accordance with the pharmaceutical recommendation.

### 2.4. Safety Evaluation

AEs were assessed using the National Cancer Institute Common Terminology Criteria for Adverse Events (CTCAE), version 5.0. Treatment was basically interrupted when any AEs of grade 3 or higher or any unacceptable drug-related AEs of grade 2 occurred. If a drug-related AE occurred, treatment was transiently interrupted until the AEs were recovered to an acceptable grade or continued a monotherapy either atezolizumab or bevacizumab, which does not correlate with AE development according to the manufacturer’s guidelines.

### 2.5. Statistical Analysis

All statistical analyses were performed using JMP statistical analysis software (JMP Pro version 14, Tokyo, Japan). PFS were calculated using the Kaplan-Meier method and analyzed using the log-rank test. All experimental data were expressed as median (range). Between-group comparisons were performed using the Mann-Whitney U test, Kruskal-Wallis test, and nonparametric analysis of variance. If one-way analysis of variance was significant, differences between individual groups were analyzed using the Fisher least significant difference test. *p* < 0.05 was considered statistically significant.

## 3. Results

### 3.1. Flow Diagram of the Study

A flow diagram of the study is shown in Figure 1. A total of 61 patients were administered atezolizumab plus bevacizumab during the study period. Among them, 4 patients were excluded because of the short observation period, and 6 patients did not finish the first radiological assessment. Finally, 51 patients were analyzed for therapeutic effects and AEs. The average observation period for all subjects after introduction of atezolizumab plus bevacizumab was 87 (26–168) days.

### 3.2. Clinical Characteristics

The characteristics of the patients are summarized in Table 1. The median age was 71 years old (37–85), and 6 patients were female. The etiology of liver diseases was non hepatitis B or C virus in 24% (*n* = 25) of patients. C-P class A and B was observed in 47 and 4 patients, respectively. ALBI grade 1, 2, and 3 was observed in 11, 39, and 1 patient, respectively. With respect to tumor characteristics, the median tumor size was 40 mm (11–132 mm) and for 88% of patients it was beyond the up-to-seven criteria. BCLC stage B and C was 24 and 27 patients, respectively. TNM stage III, IVA, and IVB was 23, 7, and 21 patients, respectively. The median alpha-fetoprotein level was 58 ng/mL (1.3–862,642 ng/mL). The median des-gamma carboxyprothrombin level was 1288 mAU/mL (18–87,529 mAU/mL). Regarding previous administration of other MTAs, 62.7% of patients (*n* = 32) had a past history of other MTAs, and 37.3% patients were MTA naïve. In the MTA-experienced patients, 22, 6 and 4 patients were second-, third-, fourth-line treatment, respectively.

### 3.3. Overall Therapeutic Outcomes of Atezolizumab plus Bevacizumab

The median number of treatment courses were 4 (1–9). The overall median PFS was 5.4 months (Figure 2A). Overall survival curve in the study is shown in Supplementary Appendix A. Overall survival did not reach the median time during the observation period (Appendix A). The radiological therapeutic response to atezolizumab plus bevacizumab was shown in Figure 2B,C. In assessment using mRECIST, complete response (CR) and partial response (PR) were observed in 4 patients (7.8%) and 14 patients (27.5%), respectively. The overall objective response rate (ORR) was 35.3% and stable disease (SD) was observed in 24 patients (51%) and progressive disease (PD) was in 7 patients (13.7%). The overall disease control rate (DCR) was 86.3%. In assessment using RECIST, CR was observed in 0 patients. PR was observed in 13 patients (25.4%). The ORR was 25.4%. SD was observed in 31 patients (60.9%). The DCR was 86.3%. PD was observed in 7 patients (13.7%). With respect to OS analysis, the patients did not reach the median survival time during the observation period.

### 3.4. Overall Safety Outcomes

The treatment related AEs are shown in Table 2. The overall incident rate of AEs at any grade and over grade 3 were 98.0% (*n* = 50) and 29.4% (*n* = 15), respectively. The most frequent AE at any grade was aspartate aminotransferase (AST) elevation, followed by alanine aminotransferase (ALT) elevation, hypertension, fatigue, and pyrexia in order. The most frequent AE over grade 3 was AST elevation, ALT elevation, and proteinuria, followed by pyrexia, hypertension, intestinal pneumoniae, and gastrointestinal perforation. Discontinuation rate due to AEs was 56.2% in the study.

### 3.5. Difference in Previous MTA History

Assessment of the therapeutic effects and safety of atezolizumab plus bevacizumab in the difference between MTA-naïve and MTA-experienced cases is shown in Figure 2. The patient characteristics in the difference of MTA previous history are shown in Appendix A. There were no significant differences between the two except in TNM stage (*p* < 0.05). The median PFS of the MTA-naïve or -experienced patients were 5.4 and 5.3 months, respectively (*p* = 0.42, Figure 3A). The radiological therapeutic response in MTA-naïve and -experienced patients is shown in Figure 3B,C. In the MTA-naïve cases, CR, PR, SD, and PD were observed in 3, 4, 1, and 11 patients, respectively, using mRECIST (Figure 3B). The ORR and DCR in the MTA-naïve cases were 36.8% and 94.5%, respectively. With respect to the MTA-experienced cases, CR, PR, SD, and PD were observed in 1, 10, 15, and 6 patients, respectively (Figure 3C). The ORR and DCR in the MTA-experienced cases was 34.3% and 71.9%, respectively. There were no significant differences in the ORR and DCR of atezolizumab plus bevacizumab between the MTA-naïve and -experienced cases (*p* = 0.96 and *p* = 0.30, respectively).

Assessment of treatment related AEs in difference of MTA-naïve or -experienced cases is shown in Table 3. There was no significant difference of frequency of AEs at any grade in MTA-naïve (94.7%) and -experienced (100%) cases (*p* = 0.15). The frequencies of AEs over grade 3 in MTA-naïve and -experienced cases were 26.3% and 31.2%, respectively. In AEs over grade 3, there was no significant difference between them (*p* = 0.70).

### 3.6. Difference in ALBI Grade

Assessment of the therapeutic effects and safety of atezolizumab plus bevacizumab in the difference between ALBI grade after initiation of treatment is shown in Figure 4. The patient characteristics in difference of ALBI grade (1 or others; 2 and 3) are shown in Appendix A. The median PFS of the ALBI grade 1 and others were 5.4 and 5.3 months, respectively (*p* = 0.98, A). The radiological therapeutic response in the ALBI grade 1 and 2 is shown in Figure 4B,C. In radiological response, CR, PR, SD, and PD was observed in 1, 5, 4, and 1 patient in ALBI grade 1 group, respectively (Figure 3B). The ORR and DCR in the ALBI grade 1 group was 54.6% and 90.9%, respectively. In ALBI others group, CR, PR, SD, and PD was observed in 3, 6, 22, and 9 patients, respectively (Figure 3C). The ORR and DCR in the ALBI others group was 30% and 85%, respectively. There was no significant difference in the therapeutic response of atezolizumab plus bevacizumab in ALBI grade (*p* = 0.18 and *p* = 0.62, respectively).

Assessment of treatment related AEs in difference of ALBI grade is shown in Table 3. There was no significant difference of frequency of AEs at any grade in ALBI 1 (100%) and ALBI others (97.5%) groups (*p* = 0.48). The frequency of AEs over grade 3 in ALBI 1 and the other cases was 18.1% and 32.5%, respectively. In AEs over grade 3, there was no significant difference between them (*p* = 0.33).

### 3.7. Therapeutic Outcomes in Child-Pugh Class B

In the present study, 4 patients were C-P class B. The number of treatment sessions was 4 (1–9) for these patients. With respect to the therapeutic response, CR, PR, SD, and PD were 1, 1, 1, and 1 patient, respectively. With regard to AEs, 4 hepatic disorders, 1 fatigue, 1 edema, 2 hypothyroidism, and 1 pyrexia at any grade were observed. There was no AE over grade 3 in C-P class B.

## 4. Discussion

The present study showed the initial experience of atezolizumab plus bevacizumab for patients with unresectable HCC in a real-world clinical practice. Atezolizumab plus bevacizumab can be administered even for the MTA-experienced cases with safety. Additionally, the study revealed that there was no significant difference of the therapeutic effects and appearance of AEs in ALBI grade. 

In the present study, we compared the therapeutic effects and safety of atezolizumab plus bevacizumab, especially in difference of previous MTA history or liver function. There were no significant differences between these groups in the therapeutic effects and safety. However, we need to assess deeply to conclude these results. In the previous MTA history, the ORR was similar between the patients with previous MTA history and naïve. However, the CR rate was 15.7% in naïve and only 3.1% in the patients with previous MTA history. With respect to ALBI grade, the ORR rate decreased from 54.6% to 30.0% in patients with other than ALBI grade 1. An incidence rate of over grade 3 AEs increased from 18.1% to 32.5%. Although there were no significant differences in the statistics, we need further accumulation of clinical experience and data in atezolizumab plus bevacizumab treatment to evaluate the efficacy and safety for the patient with previous MTA history or other than ALBI grade 1. 

In the MTAs approved through the randomized clinical trials, discrepancy of AE profiles sometimes happens in outcomes between a clinical trial and a real-world data. In the lenvatinib treatment, which is the second approved drug as first-line therapy in HCC, the frequency of general fatigue was not so high in the clinical trial [15]. However, general fatigue became an important AE in management of lenvatinib-induced AEs [16,17]. In the study, frequency of AST and ALT elevation and pyrexia was relatively high in AE profiles developed by atezolizumab plus bevacizumab compared the outcomes of the clinical trial. In the clinical trial, the frequency of liver damage was 20%. In atezolizumab plus bevacizumab, liver damage was thought to be caused by infiltration of activated T cell to the liver, especially in the patients with non-alcoholic steatohepatitis [18,19]. However, its mechanism is not fully understood. In the study, more than 50% of the enrolled patients was non-viral etiology, which might have affected the increase of liver damage. In the present study, the median periods which have appeared for AST and ALT elevation were at 19 and 21 days from the first administration of atezolizumab plus bevacizumab, respectively. Corticosteroid is recommended to treat liver damage in atezolizumab plus bevacizumab. We have used corticosteroid for patients with liver damage in atezolizumab plus bevacizumab. However, a proper management method of liver damage has not been established yet, as further evaluation is needed [20,21]. Additionally, the frequency of pyrexia was 17.9% in the clinical trial. Pyrexia is also an important AE because it can result in decreased albumin. The reason of incidence of pyrexia is not clear. Although some studies reported about the cytokine release syndrome and tumor lysis syndrome in treatment using ICI, the detailed mechanisms of pyrexia remain unclear [22,23,24]. In contrast, the frequency of hypertension was lower in the present study as compared with the clinical trial. In the clinical trial, hypertension was the most common AE. In the present study, the discontinuation rate of treatment due to AEs was 56.2%. In the clinical trial, 15.5% of the patients discontinued treatment because of the appearance of AEs. Additionally, there is a difference between our real-world data and the clinical trial. In our study, more than 60% of the patients were previously administered with other MTAs. Since all MTAs approved in the treatment of HCC can cause development of hypertension, an anti-hypertensive agent has been already administered for most of these patients. Therefore, there is a possibility that the frequency of hypertension was suppressed in the study. With respect to severe AEs, the rate of severe AEs was 29% in the study. In other hand, the rate of severe AEs in the clinical trial was 39% [8]. A shorter observation period might have affected the lower incidence rate of severe AEs in the study. Further detailed analysis is needed to clarify the mechanism developing liver damage and pyrexia in atezolizumab plus bevacizumab, and the establishment of appropriate management methods of these AEs is needed to prolong treatment period.

With regard to the overall therapeutic outcomes in atezolizumab plus bevacizumab, there was no significant difference between the clinical trial and the present study. In the clinical trial, ORR and DCR were 33.3% and 72.3%, respectively. On the other hand, ORR and DCR were 35.3% and 86.3% in the study, respectively. Although treatment was administered for patients with various tumor and patient characteristics in real-world conditions, it was valuable that atezolizumab plus bevacizumab presented with promising results. Especially, this study suggested that atezolizumab plus bevacizumab was effective even for the MTA-experienced cases. Sorafenib or lenvatinib were previously used in most of the MTA-experienced cases of the study. Some reports described that the MTA treatment induced PD-L1 expression in tumor [25,26]. The expression level of PD-L1 in tumor correlated with better therapeutic response to anti-PD-L1 treatment [27,28]. With respect to safety in the treatment of atezolizumab plus bevacizumab, there was no significant increased AEs even for the MTA-experienced cases. Further data accumulation is needed to clarify the effectiveness for the MTA-experienced cases. 

In the clinical trial of atezolizumab plus bevacizumab, the drugs were administered for patients with better liver function [8,29]. Since there was no detailed analysis in the difference of liver function in the clinical trial, the analysis in the difference of ALBI grade was performed in the study. In the present study, many cases were C-P class A. There was no significant difference between ALBI grade 1 and others (2 and 3) in the therapeutic effects and safety. In the study, atezolizumab plus bevacizumab was administered for 4 patients with C-P class B. Among these patients, all patients could be treated without the development of severe AEs until tumor progression. Since there was a possibility that atezolizumab plus bevacizumab can be administered even for patients with modest liver function, the prospective study of atezolizumab plus bevacizumab for the patients with C-P class B was needed to assess the effectiveness and safety. 

The present study has several limitations. First, the study was a retrospective study with a small sample size. Second, the observation period of treatment was relatively short. We could not evaluate OS in the study because of the short observation period. However, the present study partly revealed the effectiveness of atezolizumab plus bevacizumab in patients with HCC. Thus, further accumulation of treatment experience and analysis for a longer observation period are needed to confirm its effectiveness.

## 5. Conclusions

In conclusion, atezolizumab plus bevacizumab can be relatively safely administered to patients with previous MTA history and those with other than ALBI grade 1. However, the therapeutic effects and safety is still unclear. The present study had a short observation period; therefore, we need to evaluate the therapeutic effects of atezolizumab plus bevacizumab for unresectable HCC for a longer observation period.

## Figures and Tables

**Figure 1 cancers-13-02786-f001:**
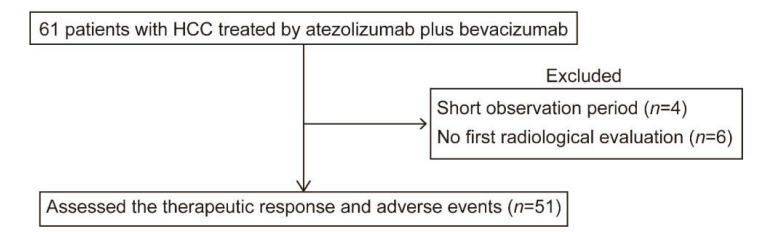
Flow diagram of the study.

**Figure 2 cancers-13-02786-f002:**
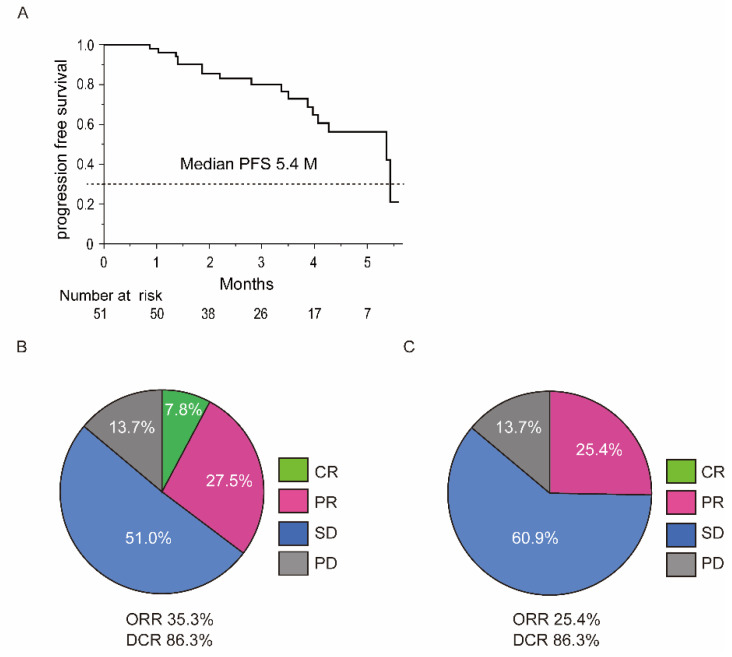
PFS and therapeutic response of atezolizumab plus bevacizumab. (**A**) PFS curve of the patients treated with atezolizumab plus bevacizumab. The median PFS was 5.4 months. (**B**) Assessment of the therapeutic response using modified RECIST. (**C**) Assessment of the therapeutic response using RECIST. Abbreviations: PFS, progression free survival.

**Figure 3 cancers-13-02786-f003:**
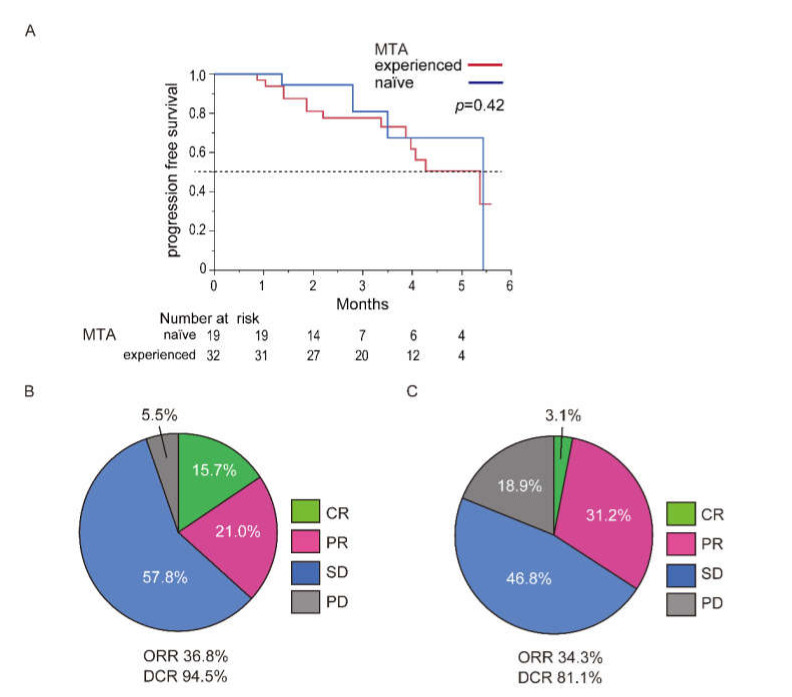
Difference of PFS and therapeutic response of atezolizumab plus bevacizumab between MTA-naïve and -experienced cases. (**A**) The PFS curves of the MTA-naïve and -experienced patients treated with atezolizumab plus bevacizumab. The blue line expresses the MTA-naïve patients. The red line expresses the MTA-experienced patients. (**B**) Assessment of the therapeutic response in MTA-naïve patients. Modified RECIST is used for assessment. (**C**) Assessment of the therapeutic response in the MTA-experienced patients. Modified RECIST is used for assessment. Abbreviations: PFS, progression free survival; MTA, molecular targeted agent.

**Figure 4 cancers-13-02786-f004:**
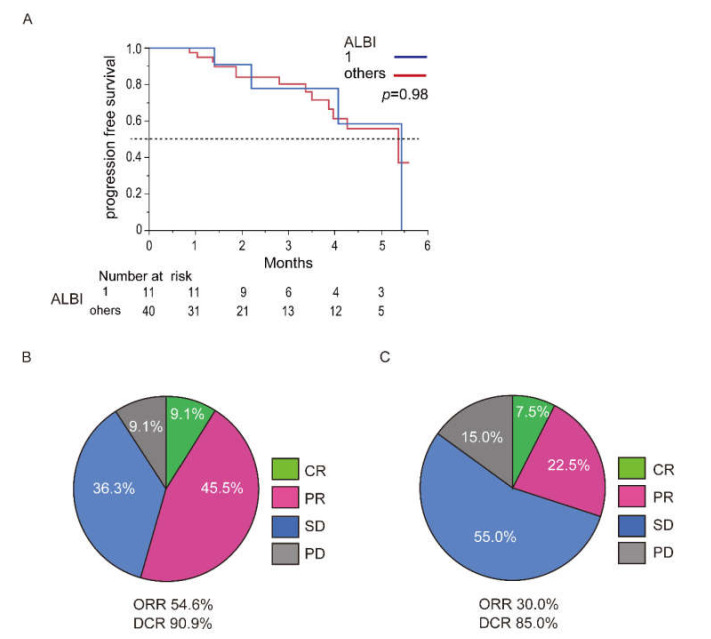
Difference of PFS and therapeutic response of atezolizumab plus bevacizumab between ALBI 1 and the others (ALBI 2 and 3). (**A**) The PFS curves of the ALBI 1 and the others (ALBI 2 and 3) patients treated with atezolizumab plus bevacizumab. The blue line expresses the ALBI 1 patient. The red line expresses the ALBI grade 2 and 3 patients. (**B**) Assessment of the therapeutic response in the ALBI 1 patients. Modified RECIST is used for assessment. (**C**) Assessment of the therapeutic response in the ALBI other patients. Modified RECIST is used for assessment. Abbreviations: PFS, progression free survival; ALBI, Albumin-bilirubin.

**Table 1 cancers-13-02786-t001:** Patient characteristics.

Characteristic	All Patients
N	51
Age (years old)	71 (37–85)
Sex (female/male)	6/45
Etiology (HBV/HCV/nonBnonC)	7/19/25
Child-Pugh score (5/6/7)	29/18/4
Child-Pugh class (A/B)	47/4
ALBI score [median (range)]	−2.39 (−3.10–−1.40)
ALBI grade (1/2/3)	11/39/1
Tumor size (mm)	40 (11–132)
Up-to-seven criteria (within/beyond)	6/45
BCLC stage (B/C)	24/27
TNM stage (III/IVA/IVB)	23/7/21
AFP (ng/mL)	58 (1.3–862,642)
DCP (mAU/mL)	1288 (18–87,529)
MTA history [(naïve/experienced (second/third/fourth)]	19/32 (22/6/4)

Data are expressed as median (range), or number. Abbreviations: ALBI score, Albumin-bilirubin score; BCLC stage, Barcelona Clinic Liver Cancer stage; AFP, α-fetoprotein; DCP, des-γ-carboxy prothrombin; MTA, molecular targeted agent.

**Table 2 cancers-13-02786-t002:** Profile of adverse events in atezolizumab plus bevacizumab.

Profile	Any Grade (*n*/%)
AST elevation	27/52.9
ALT elevation	25/49.0
Hypertension	16/31.3
Fatigue	12/23.5
Pyrexia	12/23.5
Proteinuria	10/19.6
Hypothyroidism	7/13.7
Decreased appetite	6/11.7
Hoarseness	6/11.7
Diarrhea	5/9.8
Rash	5/9.8
	Over grade 3
AST elevation	4/7.8
ALT elevation	4/7.8
Proteinuria	4/7.8
Pyrexia	2/3.9
Hypertension	2/3.9
Gastrointestinal perforation	1/1.9
Diarrhea	1/1.9
Intestinal pneumoniae	1/1.9

Abbreviations: AST, aspartate aminotransferase; ALT, alanine aminotransferase.

**Table 3 cancers-13-02786-t003:** The frequency of AE in difference of MTA history and ALBI.

**MTA History**	**MTA Naïve**	**MTA Experienced**	***p* Value**
Any grade	94.7%	100%	0.15
Over grade 3	26.3%	31.2%	0.70
ALBI grade	1	Others (2 or 3)	
Any grade	100%	97.5%	0.48
Over grade 3	18.1%	32.5%	0.33

Abbreviations: MTA, molecular targeted agent, ALBI grade, Albumin-bilirubin grade.

## Data Availability

Data is contained within the article or Supplementary Material.

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
