# Peer review of "Initial Experience of Atezolizumab Plus Bevacizumab for Unresectable Hepatocellular Carcinoma in Real-World Clinical Practice"

_cancers, 2021, doi:10.3390/cancers13112786_

Round 1
Reviewer 1 Report
Thank you for submitting your manuscript to Cancers.
This manuscript has some issues and needs to be revised.
Major problem is the definition of AEs and deficiencies in the actual treatment content and clinical course.
- What is the definition of Hepatic disorder?
The authors described that AEs were assessed using the CTCAE version 4.0 in Method. However, the item "hepatic disorder" is not defined in CTCAE version 4.0. The authors need a concrete explanation.
- The course of hepatic disorder should be described in detail.
The frequency of hepatic disorder is described as 76.4% (any grade) and 11.7% (over grade 3). It seems necessary to discuss measures and predictors for liver damage.
For all cases, it is necessary to specify how many courses were treated. Then, it is necessary to describe in detail how the interruption and dose modification were due to AE.
The most important results of this study are missing.
- The authors concluded that atezolizumab plus bevacizumab was effective and safe for patients with HCC even in the real-world setting including patients with a previous molecular targeted agent history or other than albumin-bilirubin (ALBI) grade 1. However, this conclusion is problematic.
In patients with a previous molecular targeted agent history, the ORR is similar. However, the CR rate is 15.7% vs. 3.1%, and the implications are completely different.
In patients with other than albumin-bilirubin (ALBI) grade 1, the ORR rate decreased from 54.6% to 30.0%, in addition, over grade 3 AEs increased from 18.1% to 32.5%.
Since both groups are small numbers, it may not be possible to judge by statistically significant differences alone. Moreover, it is difficult to mention Child B (7 points in all 4 cases).
Miner problems
- The fifth of authors affiliations is missing.
- The 25th citation is incomplete.
Author Response
Please see the atachment

Reviewer 2 Report
We read with interest the paper entitled “Effects and safety of atezolizumab plus bevacizumab for unresectable hepatocellular carcinoma in real-world data” by
Hideki Iwamoto et al. It is a retrospective clinical study carried out on real-life setting evaluating the effectiveness and safety of atezolizumab and bevacizumab in advanced unresectable HCC. The main message from this study is that combo therapy was effective and safe even in patients with HCC with a previous MTA history or with ALBI grade > 1.
Although clinical data coming from real life are welcoming however some drawback of this study deserve comments:
- This is a retrospective study carried out on a small series of patients. The authors should specify whether they adopted a sort of intention-to-treat recruitment or not and, if not, how many patients were offered the treatment and how many of them refused or were unable to start it.
- The authors could not assess the overall survival (OS). This is a relevant limit of the study. OS indeed, is an important outcome (and a primary end point in registration studies) to evaluate the effectiveness of treatment. The median observation period in the study was only 81 days which is an interval inadequate to calculate OS. A longer observation period would have allowed the assessment of this outcome. The lack of information on OS greatly reduces the robustness of authors conclusions Likely, the shorter median PFS observed in the study (5.4 months) compared to that observed in the NEJM paper by Finn et al (6.8 months) may depend on the inadequate observation period.
- Another point deserving comment is the lower rate of serious adverse events observed in the study (29%) as compared to that reported in RCT by Finn et al (39%). The authors should comment on this discrepancy.
In summary, the retrospective design, the small number of patients and the short observation period all together greatly reduce strength of clinical message of the study. However, it may represent an early contribution on the effectiveness of this MTA combo treatment of patients with advanced HCC in the real clinical practice deserving confirmation in larger series.
Author Response
Please see the atachment.

Round 2
Reviewer 1 Report
The authors responded in good faith to the reviewers' comments. In the revised edition, the problems pointed out in the original edition have been resolved and the quality of the treatise has improved. This article is expected to provide useful information to readers of cancers.
Reviewer 2 Report
No further clarifications are required